# Pointer States and Quantum Darwinism with Two-Body Interactions

**DOI:** 10.3390/e25121573

**Published:** 2023-11-22

**Authors:** Paul Duruisseau, Akram Touil, Sebastian Deffner

**Affiliations:** 1ENS Paris-Saclay, 91190 Gif-sur-Yvette, France; paul.duruisseau@ens-paris-saclay.fr; 2Theoretical Division, Los Alamos National Laboratory, Los Alamos, NM 87545, USA; atouil@lanl.gov; 3Center for Nonlinear Studies, Los Alamos National Laboratory, Los Alamos, NM 87545, USA; 4Department of Physics, University of Maryland, Baltimore County, Baltimore, MD 21250, USA

**Keywords:** quantum Darwinism, decoherence

## Abstract

Quantum Darwinism explains the emergence of classical objectivity within a quantum universe. However, to date, most research on quantum Darwinism has focused on specific models and their stationary properties. To further our understanding of the quantum-to-classical transition, it appears desirable to identify the general criteria a Hamiltonian has to fulfill to support classical reality. To this end, we categorize all *N*-qubit models with two-body interactions, and show that only those with separable interaction of the system and environment can support a pointer basis. We further demonstrate that “perfect” quantum Darwinism can only emerge if there are no intra-environmental interactions. Our analysis is complemented by solving the ensuing dynamics. We find that in systems exhibiting information scrambling, the dynamical emergence of classical objectivity directly competes with the non-local spread of quantum correlations. Our rigorous findings are illustrated through the numerical analysis of four representative models.

## 1. Introduction

We live in a quantum universe, yet our everyday reality is well-described by classical physics. Hence, this leads to the obvious question: where do all the quantum information and correlations hide? The quantum nature of our universe is captured by its ability to be in a superposition of classically allowed states. The transition from quantum to classical is a two-step process. The first (necessary but not sufficient) step is the destruction of quantum superpositions, i.e., the destruction of all interference phenomena. The theory of decoherence teaches us that it is the interaction between the quantum system and its environment that causes this phenomenon [1]. The destruction of quantum superpositions identifies a privileged and unique quantum basis. The elements of this basis are called pointer states [2,3,4]. Decoherence theory then concludes that for typical Hamiltonians, any quantum superposition expressed in this basis decomposes into a classical mixture under the effect of environmental interaction. Thus, pointer states are precisely the only quantum states that remain stable under this interaction.

Quantum Darwinism [5,6,7,8,9,10,11,12,13] builds on the decoherence theory and goes a step further, approaching the problem from the point of view of quantum information theory. An outside observer has no direct access to a system of interest S, but rather the environment E acts as a communication channel. Since any real environment is tremendously large, “observing” S actually means that an observer intercepts only a small, possibly even tiny fragment F of E, and then reconstructs the state of S from the information carried by F.

If the constituents of the environment do not interact, such as, for instance, photons [14,15], then the information about S is accessible by *local* measurements on E. However, reality is a little more complicated, and in general, the constituents of E do interact. Such intra-environmental interactions lead to the build-up of non-local correlations, which is the root cause of information scrambling [16,17,18,19,20,21]. Thus, an observer has to access a macroscopic fraction of E to reconstruct unambiguous information about S.

Despite the significant attention that scrambling dynamics has received in the literature [16,17,18,19,20,21,22,23], curiously, little is known about the quantum-to-classical transition in the context of scrambling. Only recently have several studies begun to unravel the interplay between decoherence and scrambling [23,24,25,26,27,28,29,30,31].

More directly relevant to our present work is Ref. [32], which analyzed a specific model where intra-environmental interactions scramble the information encoded in different fragments F. This scenario, scrambling only in E and not in S, makes it easier to highlight the competition between the local transfer of information from S to each degree of freedom of E, and the scrambling of information between the different F of E due to their interactions.

Current research in quantum Darwinism [13] is driven by the analysis of increasingly complex model systems. However, the focus has remained on particular qubit models [32,33,34,35,36,37] since their dynamics are tractable. Despite—or rather because of—the continued progress in our understanding, it seems desirable to elucidate the general properties of Hamiltonians that support the emergence of quantum Darwinism. More precisely, it is beneficial to categorize all possible interacting many-body Hamiltonians into classes that support a pointer basis for S, and to determine which subclasses of these will further exhibit the emergence of classical objectivity. Such a classification will also reveal whether and under what circumstances quantum Darwinism can emerge in the presence of scrambling dynamics.

In the present work, we consider a qubit of interest S, which interacts with an environment, E, also comprised of qubits. Hence, the scrambling of information may only occur in E, but not in S. Further, for the sake of simplicity, we restrict ourselves to arbitrary two-body interactions.

In the first part of our analysis, we show that the existence of a pointer basis for S imposes a specific structure for the total Hamiltonian describing the evolution of the universe, S⊗E. In fact, we will see that a pointer basis for S exists for any interactions within E, yet S may only interact with all fragments of E, identically, cf. Figure 1. The second part of the analysis is then focused on the dynamics induced by such Hamiltonians that support a pointer basis. We find that the efficiency of the information transfer between S and E is governed by the statistics of the interaction terms. The *average* information transfer is irreversible if and only if the support of the coupling coefficients is continuous, and the “speed of communication” is determined by the shape of the distribution of the interaction coefficients. Note that we average the information transfer over many instances of models. Our general findings are illustrated with four models that correspond to a variety of situations, including scrambling or no scrambling, pointer basis or no pointer basis, quantum Darwinism or no quantum Darwinism.

## 2. Structure of the Hamiltonian

We start by defining the problem in mathematically rigorous terms. Consider a set of (N+1) qubits, where the 0th qubit is the system S. Hence, the environment E is comprised of *N* qubits. For the sake of simplicity, we further restrict ourselves to two-body interaction models. The most general Hamiltonian corresponding to this scenario then reads
(1)$$H=\sum_{i,j}^{}\sum_{\alpha ,\beta }^{}J_{ij}^{\alpha \beta }\sigma_{i}^{\alpha }⊗\sigma_{j}^{\beta }+\sum_{i}^{}\vec{B_{i}}\cdot \vec{\sigma_{i}},$$
where α and β take the values *x*, *y*, and *z*, corresponding to the Pauli matrices $\sigma^{x}$, $\sigma^{y}$, and $\sigma^{z}$. Indices *i* and *j* count the qubits, with i,j=0 for S and i,j≥1 for E, and correspondingly, $I_{j}$ is the identity acting on the *j*th qubit. Finally, $J_{ij}$ and $\vec{B_{i}}$ are real coefficients, which in the following we will choose to be random variables.

### 2.1. Existence of a Pointer Basis

The natural question now arises: what conditions must $J_{ij}$ and $\vec{B_{i}}$ fulfill in order for *H* in Equation (Equation 1) to support a pointer basis for S? Pointer states are *the* particular states of S that are stable under the interaction with E [2,3,4]. Formally, these states can be identified in the following way: $|\psi_{S}\rangle \in S$ is a pointer state of S if and only if for any $|\psi_{E}\rangle \in E$, an initial product state $|\psi_{S}\rangle ⊗|\psi_{E}\rangle$ evolves under *H* (Equation 1) to remain within an epsilon ball around the product state $|\psi_{S}\rangle ⊗|\psi_{E}\left(t\right)\rangle$. In other words, the reduced state $|\psi_{S}\rangle$ remains pure under the evolution of the total Hamiltonian.

It will prove convenient to separate the total Hamiltonian into terms corresponding to S, E, and their interactions. Hence, we write
(2)$$H=H_{S}⊗I_{E}+I_{S}⊗H_{E}+H_{\mathrm{SE}}\;.$$
Compared with Equation (Equation 1), we identify the system Hamiltonian as
(3)$$H_{S}=\vec{B_{0}}\cdot \vec{\sigma_{0}}\;,$$
whereas, for the environment, we have the following equation
(4)$$H_{E}=\sum_{1\leq i<j\leq N}^{}\sum_{\alpha ,\beta }^{}J_{ij}^{\alpha \beta }\sigma_{i}^{\alpha }⊗\sigma_{j}^{\beta }+\sum_{i=1}^{N}\vec{B_{i}}\cdot \vec{\sigma_{i}}\;.$$
Notice that the first term in Equation (Equation 4) describes the intra-environmental interactions. The interaction between S and E is given by
(5)$$H_{\mathrm{SE}}=\sum_{\alpha }^{}\sigma_{0}^{\alpha }⊗\sum_{j=1}^{N}\sum_{\beta }^{}J_{0j}^{\alpha \beta }\sigma_{j}^{\beta }\;.$$
From this separation of terms, it becomes immediately obvious that the pointer basis for S can only exist if certain necessary and sufficient conditions for the interaction term $H_{\mathrm{SE}}$ are fulfilled.

These conditions become particularly intuitive by considering the original motivation for *pointer* states. These states are not only immune to the dynamics induced by the interaction with the environment but can also be thought of as states that correspond to the *pointer* of a measurement apparatus. Mathematically, such an apparatus is described by the pointer observable given by the following equation:(6)$$A\equiv A_{S}⊗I_{E}\;.$$
By definition of pointer states, the pointer observable *A* commutes with the total Hamiltonian (Equation 1) and, hence, *A* and *H* share an eigenbasis. Due to the form of *A* the corresponding eigenstates can be written in tensor-product form $|S_{i}\rangle ⊗|E_{j}\rangle$ with $|S_{i}\rangle \in S$ and $|E_{j}\rangle \in E$.

Correspondingly, we can factorize the time-evolution operator as
(7)$$U=\sum_{i}|S_{i}\rangle \;\langle S_{i}|⊗\exp\left(-i/\hbar \;H_{i}\;t\right)$$
where $H_{i}$ acts only on E. Now, choosing any (reduced) eigenstate of *A* as the initial state of S, $|\psi_{S}\rangle =|S_{i}\rangle$, the product state $|\psi_{S}\rangle ⊗|\psi_{E}\rangle$ evolves by remaining in product form $|\psi_{S}\rangle ⊗|\psi_{E}\left(t\right)\rangle$.

In Appendix A, we show that by enforcing the commutation relation $\left[A,\;H\right]=0$, we have that any Hamiltonian (Equation 4) supporting a pointer basis for S has to be of the form
(8)$$H=H_{S}⊗I_{E}+I_{S}⊗H_{E}+H_{S}⊗\sum_{i=1}^{N}h_{i},$$
where $H_{S}$ is the system Hamiltonian (Equation 3), and $h_{i}$ denotes arbitrary traceless Hermitian operators acting on the *i*th qubit of E.

In conclusion, we show that any model of interacting qubits that supports a pointer basis for a system qubit S may have, at most, a separable interaction term, $H_{\mathrm{SE}}$. Moreover, these interaction terms have to be factorizable into the system Hamiltonian $H_{S}$ and traceless terms acting on the environmental qubits E. It is important to emphasize that no additional conditions are required pertaining to, for instance, the intra-environmental interactions. Schematically, our findings are illustrated in Figure 1.

### 2.2. Further Conditions for Quantum Darwinism

It was demonstrated in Ref. [11] that only a special structure of states is compatible with the emergence of quantum Darwinism. These states are of the *singly branching form* [12,38], which are the only states to support epsilon quantum correlations as measured by quantum discord [39].

Singly branching states are pointer states of S correlated with the environment states in the special form,
(9)$$|\psi (t)\rangle =\alpha_{0}|0\rangle \underset{i\in E}{⨂}|0_{i}\left(t\right)\rangle +\beta_{0}|1\rangle \underset{i\in E}{⨂}|1_{i}\left(t\right)\rangle \;.$$
It is easy to see that such a singly branching form can emerge if and only if there are no intra-environmental interactions.

Thus, we conclude that quantum Darwinism can only be supported by Hamiltonians with separable interactions between S and E, cf. Equation (Equation 8), and no intra-environmental interactions, i.e., $J_{ij}^{\alpha \beta }=0$ in Equation (Equation 4). The remaining question now is whether all such Hamiltonians provide so-called good decoherence, which makes their corresponding E good channels for information transfer.

## 3. Coefficients of the Hamiltonian

To analyze the *dynamical* emergence of quantum Darwinism, we now solve for the average dynamics under an arbitrary, random Hamiltonian for which the system S has a pointer basis of the single branching form (Equation 9). We find that the efficiency of information transfer within E is governed by the randomness of the interaction coefficients.

### 3.1. Solving the Dynamics

To this end, consider an arbitrary Hamiltonian of the form (Equation 8), where we further enforce vanishing intra-environmental interactions $J_{ij}^{\alpha \beta }=0$. Note that in Equation (Equation 8), $h_{i}$ is Hermitian and traceless. Hence, we can write equivalently (and without loss of generality)
(10)$$H=\sigma_{0}^{z}⊗\sum_{i=1}^{N}B_{i}\;\sigma_{i}^{z},$$
where $B_{i}$ denotes real random variables.

We are now interested in the dynamics induced by Equation (Equation 10), and we choose an arbitrary separable initial condition. Therefore, we write
(11)$$|\psi_{0}\rangle =\left(\alpha_{0}|0\rangle +\beta_{0}|1\rangle \right)\;\overset{N}{\underset{i=1}{⨂}}\left(\alpha_{i}|0_{i}\rangle +\beta_{i}|1_{i}\rangle \right),$$
where $\alpha_{i}$ and $\beta_{i}$ are arbitrary, complex coefficients. Evolving this $|\psi_{0}\rangle$ under the corresponding Schrödinger equation, $i\partial_{t}|\psi \rangle =H|\psi \rangle$, we obtain the time-dependent solution,
(12)$$|\psi (t)\rangle =\alpha_{0}|0\rangle \overset{}{\underset{i\in E}{⨂}}|0_{i}\left(t\right)\rangle +\beta_{0}|1\rangle \overset{}{\underset{i\in E}{⨂}}|1_{i}\left(t\right)\rangle ,$$
where we introduce
(13)$$\begin{array}{cc} |0_{i}\left(t\right)\rangle & =\alpha_{i}\;\exp\left(iB_{i}t\right)|0\rangle +\beta_{i}\;\exp\left(-iB_{i}t\right)|1\rangle \\ |1_{i}\left(t\right)\rangle & =\alpha_{i}\;\exp\left(-iB_{i}t\right)|0\rangle +\beta_{i}\;\exp\left(iB_{i}t\right)|1\rangle \;. \end{array}$$

As usual, the reduced density matrix of S is given by tracing out E
(14)$$\rho_{S}\left(t\right)={\mathrm{tr}}_{E}\left\{|\psi (t)\rangle \langle \psi (t)|\right\}\;.$$
The corresponding decoherence factor [32] is given by the amplitude of the off-diagonal coefficients of the reduced density matrix $\rho_{S}$ in the basis $\left\{|0\rangle ,|1\rangle \right\}$. We have
(15)$$\langle {\left|\langle 1_{i}\left(t\right)|0_{i}\left(t\right)\rangle \right|}^{2}\rangle \equiv \langle {\left|\Gamma_{i}\left(t\right)\right|}^{2}\rangle \;,$$
and since $B_{i}$ is stochastically independent, we can write
(16)$$\langle {\left|\prod \Gamma_{i}\left(t\right)\right|}^{2}\rangle =\prod \langle {\left|\Gamma_{i}\left(t\right)\right|}^{2}\rangle .$$
It is easy to see that we have
(17)$$\Gamma_{i}\left(t\right)={\left|\alpha_{i}\right|}^{2}\;\exp\left(-2iB_{i}t\right)+{\left|\beta_{i}\right|}^{2}\;\exp\left(2iB_{i}t\right)\;.$$
For the random $B_{i}$, it is, however, more instructive to compute the decoherence factors averaged over all possible values for $B_{i}$. By further denoting the probability density function of $B_{i}$ as $P\left(B_{i}\in \left[x,x+dx\right]\right)=f_{i}\left(x\right)dx$, we show in Appendix B that we obtain
(18)$$\begin{array}{cc} \langle {\left|\Gamma_{i}\left(t\right)\right|}^{2}\rangle & ={\left|\alpha_{i}\right|}^{4}+{\left|\beta_{i}\right|}^{4}\\ & +2{\left|\alpha_{i}\right|}^{2}{\left|\beta_{i}\right|}^{2}\left|\tilde{f_{i}}\left(4t\right)\right|\cos\left(\arg\left(\tilde{f_{i}}\left(4t\right)\right)\right), \end{array}$$
where $\tilde{f_{i}}$ is the characteristic function of $B_{i}$:(19)$$\tilde{f_{i}}\left(k\right)=\int_{-\infty }^{+\infty }dx\;f_{i}\left(x\right)\;\exp\left(ikx\right)\;.$$

In conclusion, we derived an analytic expression for the average decoherence factor, which governs the rate at which information about S is communicated through the environment E.

### 3.2. Rate and Irreversibility of Information Transfer

Equation (Equation 18) demonstrates the relationship between the emergence of classicality and the randomness of system–environment interactions. Indeed, the probability distributions, $f_{i}$, of the couplings, $B_{i}$, between S and E play a central role in determining the rate of information transfer. We can see that the decoherence factors decrease rapidly if $\tilde{f_{i}}$ rapidly decreases. Since $\tilde{f_{i}}$ is the Fourier transform of the probability distribution, $f_{i}$, the order of differentiability of $f_{i}$ gives us the order of decay of $\tilde{f_{i}}$, while the smallest characteristic length in the distribution, $f_{i}$, gives us the inverse of the characteristic time of decay of $\tilde{f_{i}}$.

Furthermore, if the support of $B_{i}$ is discrete and finite, then the characteristic function $\tilde{f_{i}}\left(k\right)$ is a periodic (or quasi-periodic) function and, therefore, does not converge to 0. Hence, having continuous support for $f_{i}$ is essential for the emergence of truly classical behavior (as averaged over ensembles of models). In fact, if the $f_{i}$ distribution has continuous support, then the information is transferred *irreversibly*. In this case, $f_{i}$ is integrable
(20)$$\int_{-\infty }^{+\infty }dx\;\left|f_{i}\left(x\right)\right|=\int_{-\infty }^{+\infty }dx\;f_{i}\left(x\right)=1<\infty ,$$
and, thus, by virtue of the Riemann–Lebesgue Lemma,
(21)$$\left|\tilde{f_{i}}\left(k\right)\right|\underset{k\to \infty }{\to }0\;\mathrm{and}\;\mathrm{thus}\;\langle {\left|\Gamma_{i}\left(t\right)\right|}^{2}\rangle \underset{t\to \infty }{\to }\epsilon_{i}<1.$$
where $\epsilon_{i}$ depends on the initial state.

Finally, we note that a perfect record of the information about S in the *i*th qubit corresponds to $\Gamma_{i}=0$. This is typically not the case. However, as the values of $\langle {\left|\Gamma_{i}\left(t\right)\right|}^{2}\rangle$ become strictly less than one, Equation (Equation 16) shows that $|0\rangle {⨂}_{i\in F}^{}|0_{i}\left(t\right)\rangle$ and $|1\rangle {⨂}_{i\in F}^{}|1_{i}\rangle$ become orthogonal on average for a sufficiently large fragment F. Thus, a small set of qubits in the environment is sufficient to obtain an almost complete record of the state of S.

### 3.3. Quantum Darwinism—The Classical Plateau

The hallmark result of quantum Darwinism is the emergence of the “classical plateau” [5,6,7,8,9,10,11,12,13], cf. Figure 2. This classical plateau is a consequence of redundant encoding of the same information in E.

To this end, consider again a fragment F of E. If any F carries the same information about S, then any two observers accessing different F learn exactly the same information about S. The amount of information that a fragment F of E contains about the system S can be quantified with mutual information I(S:F), defined as
(22)$$I\left(S:F\right)=S_{S}+S_{F}-S_{\mathrm{SF}}\;,$$
where $S\left(\rho \right)=-\mathrm{tr}\left\{\rho \log(\rho )\right\}$. The maximal classical information that can be accessed by any observer is upper-bounded by the Holevo quantity [40,41]
(23)$$\chi \left(S:\overset{ˇ}{F}\right)=S_{S}-S_{S|\overset{ˇ}{F}}$$
where $S_{S|\overset{ˇ}{F}}$ is the conditional von Neumann entropy defined as the minimal von Neumann entropy of S obtained after a measurement on F.

The difference between the mutual information, I(S:F), and the Holevo quantity, $\chi (S:\overset{ˇ}{F})$, has been termed quantum discord [39],
(24)$$D\left(S:\overset{ˇ}{F}\right)\equiv I\left(S:F\right)-\chi \left(S:\overset{ˇ}{F}\right)\geq 0\;.$$
Quantum discord measures the genuinely quantum information encoded in F.

For each F and its complement, $\bar{F}=E\setminus F$, we define the corresponding decoherence factors
(25)$$\Gamma_{F}=\prod_{i\in F}^{}\Gamma_{i}\;\mathrm{and}\;\Gamma_{\bar{F}}=\prod_{i\notin F}^{}\Gamma_{i}.$$
With these definitions, one can then show that for sufficiently small decoherence factors [32], we have
(26)$$I\left(S:F\right)≃S_{max}-\frac{\xi ({\left|\alpha_{0}\right|}^{2})}{2}\left[{\left|\Gamma \right|}^{2}+{\left|\Gamma_{F}\right|}^{2}-{\left|\Gamma_{\bar{F}}\right|}^{2}\right],$$
where $S_{max}=-{\left|\alpha_{0}\right|}^{2}\log\left({\left|\alpha_{0}\right|}^{2}\right)-\left(1-{\left|\alpha_{0}\right|}^{2}\right)\log\left(1-{\left|\alpha_{0}\right|}^{2}\right)$, which represents the maximal value of the von Neumann entropy of S. Correspondingly, for the Holevo quantity, we obtain the following (see Appendix C):(27)$$\chi \left(S:\overset{ˇ}{F}\right)≃S_{max}-\frac{\xi ({\left|\alpha_{0}\right|}^{2})}{2}{\left|\Gamma_{F}\right|}^{2}\;,$$
which is fully consistent with earlier findings [42].

Finally, in the limit of long times, when t≫1, and for sufficiently smooth $f_{i}$ (Equation 21), we obtain the following asymptotic expression for mutual information
(28)$$\begin{array}{c} I\left(S:F\right)≃S_{max}-\frac{\xi ({\left|\alpha_{0}\right|}^{2})}{2}\left[\prod_{i\in E}^{}\epsilon_{i}+\prod_{i\in F}^{}\epsilon_{i}-\prod_{i\notin F}^{}\epsilon_{i}\right]\;, \end{array}$$
and the Holevo quantity
(29)$$\chi \left(S:\overset{ˇ}{F}\right)≃S_{max}-\frac{\xi ({\left|\alpha_{0}\right|}^{2})}{2}\prod_{i\in F}^{}\epsilon_{i}\;.$$
Further, averaging over every possible separable initial state, we have
(30)$$I{\left(S:n\right)}_{\infty }≃S_{max}-\frac{\xi ({\left|\alpha_{0}\right|}^{2})}{2}\;\left[{\bar{\epsilon }}^{N}+{\bar{\epsilon }}^{n}-{\bar{\epsilon }}^{N-n}\right]$$
and
(31)$$\chi {\left(S:n\right)}_{\infty }≃S_{max}-\frac{\xi ({\left|\alpha_{0}\right|}^{2})}{2}\;{\bar{\epsilon }}^{n}\;,$$
where *n* is the size of F and $\bar{\epsilon }=2/3$ (see Appendix C).

These results are depicted in Figure 2 for N=50 environmental qubits. Both the mutual information and the Holevo quantity display a sharp initial increase with the growing size of the fragment, *n*, as larger fragments yield more data about S. This initial rise is followed by the classical plateau.

## 4. Representative Examples

We conclude the analysis with the numerical solutions of four representative examples. To support quantum Darwinism, a Hamiltonian must obey the following conditions: the existence of a pointer basis, continuous support, and no intra-environment interactions. Our first example exhibits these three conditions, and for each following example, we successively remove one of these conditions, cf. Table 1.

### 4.1. Continuous Parallel Decoherence Interaction

The first model has a pointer basis, random coupling coefficients with a continuous spectrum, and does not exhibit scrambling in E. The corresponding Hamiltonian reads
(32)$$H_{\mathrm{CPDI}}=\sigma_{0}^{z}⊗\sum_{i=1}^{N}B_{i}\sigma_{i}^{z}$$
where $B_{i}$ denotes independent random variables, drawn uniformly from $B_{i}\in \left[-1,1\right]$. For specificity, we call this the *continuous parallel decoherence interaction (CPDI)* model.

In Figure 3 a, we plot the resulting mutual information (Equation 22), as a function of the fragment size, which rapidly converges toward the asymptotic expression (Equation 28). Note the distinct classical plateau indicative of quantum Darwinism. Moreover, we observe the relaxation of S into its stationary pointer states over a typical time τ≃1/4, at which point, the information transfer becomes irreversible.

### 4.2. Discrete Parallel Decoherence Interaction

Our second example is called *discrete parallel decoherence interaction (DPDI)*. The corresponding Hamiltonian is
(33)$$H_{\mathrm{DPDI}}=\sigma_{0}^{z}⊗\sum_{i=1}^{N}B_{i}\sigma_{i}^{z}$$
where $B_{i}$ denotes, again, independent random variables. However, in contrast to the continuous case in Equation (Equation 32), $B_{i}$ is now drawn uniformly from the *discrete* set $B_{i}\in \left\{-1,-0.5,0.5,1\right\}$.

In Figure 3b, we depict the resulting mutual information. As expected, we observe that the classical plateau appears and disappears periodically and, hence, the information transfer is no longer irreversible. At instants at which the plateau completely vanishes, I(S:F) is linear in the fragment size. This indicates that the information about S encoded in E is distributed throughout the entire environment (no redundancy).

### 4.3. Continuous Orthogonal Decoherence Interaction

As the next example, we consider the *continuous orthogonal decoherence interaction (CODI)*, which refers to our first model (Equation 32) but with an added external field. This additional term is chosen to be not parallel to the interaction between S and E and, hence, CODI does not support a pointer basis. The Hamiltonian reads
(34)$$H_{\mathrm{CODI}}=\sigma_{0}^{y}⊗I_{E}+\sigma_{0}^{z}⊗\sum_{i=1}^{N}B_{i}\sigma_{i}^{z}.$$
where, as before, $B_{i}$ denotes independent random variables, drawn uniformly from $B_{i}\in \left[-1,1\right]$.

In this model, information about the observable $\sigma_{0}^{z}$ can be registered in E, but the eigenstates of this observable are not stable and, hence, not classically objective. This observation is further supported by Figure 3c, which does not exhibit any classical plateau form. Moreover, at all instants, I(S:F) is a linear function of the size of F, which is a consequence of the complete absence of any redundancy.

### 4.4. Continuous Parallel Decoherence Interaction with Scrambling

As a final example, we again consider Equation (Equation 32) but now design E to exhibit the scrambling dynamics. Accordingly, this model is called the continuous parallel decoherence interaction with scrambling (CPDI-S), and the Hamiltonian becomes
(35)$$H_{\mathrm{CPDI}-S}=\sigma_{0}^{z}⊗\sum_{i=1}^{N}B_{i}\sigma_{i}^{z}+\sum_{1\leq i<j\leq N}^{}J_{ij}\sigma_{i}^{z}⊗\sigma_{j}^{z}$$
where, again, $B_{i}$ denotes independent random variables, drawn uniformly from $B_{i}\in \left[-1,1\right]$, and $J_{ij}$ denotes independent random variables, drawn uniformly from $J_{ij}\in \left[-0.03,0.03\right]$.

As Figure 3d shows, a classical plateau rapidly emerges over a time scale of $\tau_{\mathrm{SE}}≃1/4$. However, this plateau quickly “disperses” as the quantum information becomes non-local due to scrambling in E. We refer to Ref. [32] for a more detailed analysis of this particular model. Furthermore, it is interesting to note that Equation (Equation 35) is another example that demonstrates the competition of decoherence and scrambling as a “sink for quantum information” as analyzed by (some of) us in Ref. [25].

## 5. Concluding Remarks

In the present work, we determined the set of qubit models that support the emergence of classicality. In particular, we established a classification of two-body interaction models based on the structure of the Hamiltonian and the nature of its coefficients.

The existence of an “exact” pointer basis for the qubit S requires the interaction Hamiltonian to be separable between S and its environment E, such that the part acting on S is proportional to the self-Hamiltonian of S. We call that type of structure a *parallel decoherence interaction*. Furthermore, without any intra-environment interactions, this Hamiltonian structure leads to a branching state structure, the only one compatible with quantum Darwinism [11].

Furthermore, intra-environment interactions can lead to information scrambling in E, which deteriorates the branching structure. In such situations, the state of S is still a classical mixture, but this classicality is hidden from an outside observer who must measure a non-local part of E in order to recover almost all the information about S. This indicates a clear competition between the emergence of classical objectivity and scrambling dynamics.

The conceptual notions and the gained insight from our work may open up the door for further inquiries, such as the study of k−body interactions. In particular, there is every reason to believe that this type of interaction leads to a non-local information transfer. Indeed, for such interactions, the information about S is directly encoded in E by the entanglement of S with fragments F of size *k*. This results in a lower redundancy of information. However, the analytical analysis of the dynamics is much more involved than the present two-body interaction case, which is why we leave k−body interactions for future work. 

## Figures and Tables

**Figure 1 entropy-25-01573-f001:**
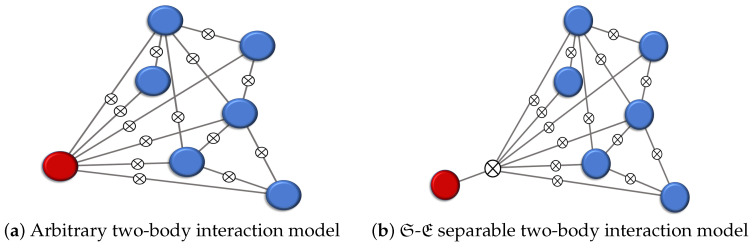
Two-body interactions between S (red) and E (blue). Lines with ⊗ depict interaction terms. (**a**) Most general scenario in which S might separately interact with all qubits in E. (**b**) Scenario with separable interaction, i.e., S interacts with E through a global tensor product structure; this scenario supports a pointer basis for S.

**Figure 2 entropy-25-01573-f002:**
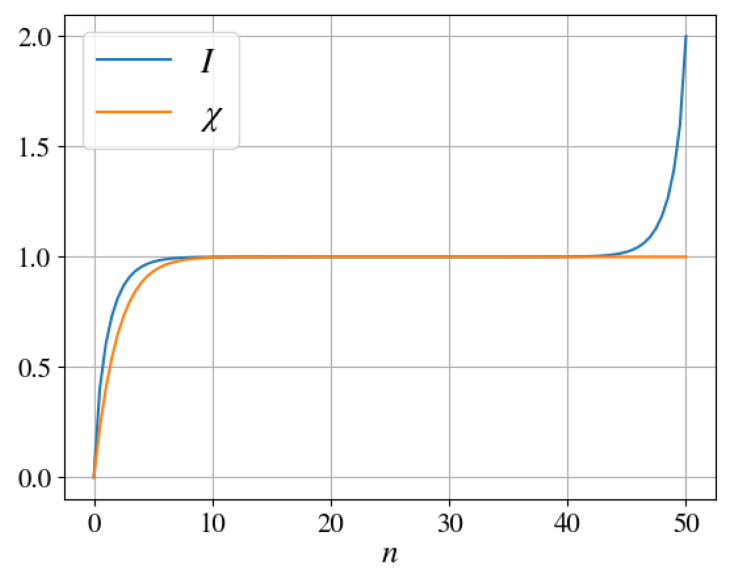
Asymptotic mutual information (Equation 28) and Holevo quantity (Equation 29) as functions of the fragment size. The classical plateau corresponds to the value $S_{max}=1$.

**Figure 3 entropy-25-01573-f003:**
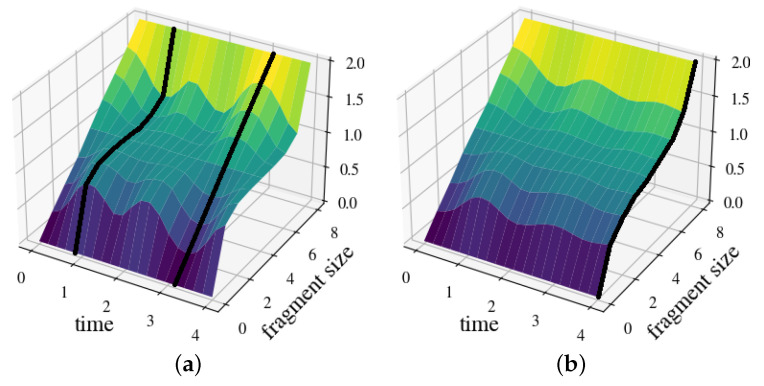
Mutual information I(S:F) as a function of time and fragment size for an arbitrary separable initial state of 9 qubits (N=8), divided by the von Neumann entropy of S (averaged over $10^{2}$ realizations). (**a**) CPDI (Equation 32): irreversible information transfer and classical objectivity. (**b**) DPDI (Equation 33): non-irreversible (periodic) information transfer. (**c**) CODI (Equation 34): no local information transfer or redundancy. (**d**) CPDI-S (Equation 35): competition of emergence of classical objectivity and scrambling in E.

**Table 1 entropy-25-01573-t001:** Model classification.

	Pointer Basis	Continuous Support	No Scrambling
CPDI (Equation 32)	✓	✓	✓
DPDI (Equation 33)	✓	×	✓
CODI (Equation 34)	×	✓	✓
CPDI-S (Equation 35)	✓	✓	×

## Data Availability

Data are contained within the article.

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
