# Peer review of "Pointer States and Quantum Darwinism with Two-Body Interactions"

_entropy, 2023, doi:10.3390/e25121573_

Round 1

Reviewer 1 Report

Comments and Suggestions for Authors

This is an excellent, well-written paper on an important topic, with significant results that greatly contribute to our theoretical understanding of the quantum-classical transition. I wholeheartedly recommend publication in present form.

The paper provides detailed and clear calculations that elucidate the conditions that must be imposed on the form of system-environment Hamiltonians in order to enable certain features of the quantum-classical transition, namely, the existence of a pointer basis, the emergence of quantum Darwinism, and the rate with which information about the system is encoded in the environment.

This paper builds on preliminary insights from prior work (e.g, Ref. [32] quoted by the authors), but presents a much more rigorous, general take on the different aspects of the quantum-classical transition and their relation to the structure of the system-environment Hamiltonian, especially when it comes to separability and to interactions between the constituents of the environment. As an additional bonus, the authors also numerically study four explicit models.

The clarity of the writing style and the editing of the paper are superb. The figures are excellent; my only very minor suggestion would be to add an explanatory sentence to Figure 1, to make clear the difference between (a) and (b) and its implications. I found just one tiny editing issue: In line 89, it should read "Due to the form..." (the word "the" is missing).

In summary, this is an important and well-written paper, and I recommend its publication in Entropy.

Author Response

We would like to thank the referee for their strong support, kind words, and very positive recommendation. In our revisions, we have addressed both minor issues pointed out in the report.

Reviewer 2 Report

Comments and Suggestions for Authors

In this manuscript the authors consider the ability of qubit models with arbitrary two-qubit interactions to support a mathematically idealized form of quantum Darwinism (QD) characterized by there being a basis of pointer states for the system (a particular qubit) that remain exactly pure under time evolution. They observe that, for qubit systems, this idealized QD requires an interaction Hamiltonian that factorizes. They then consider a further idealized ("perfect") form of QD producing so called "single-branching states", and compute the amount of information transferred to the environment after averaging over some ensemble of coupling coefficients. They plot this for some specific models that do and do not fulfill the idealizations. The usefulness of these results are modest because the authors rely heavily on exact idealizations, and because they consider ensemble averages of information transfer where the ensemble has unclear physical interpretation.

I previously reviewed this manuscript for another journal. I am glad to see that the authors have resolved some of the issues that I had raised, although several remain. I divide the remaining issues into "necessary" ones, which I ask the authors to either fix or explain in their reply, and "optional" ones, which the authors can choose to ignore.

=== Necessary issues ===

(*) The authors write

> In fact, if the f_i distribution is continuous, then the information is transferred irreversibly.

Here they appear to be confusing continuity of the probability distribution over coefficients with an infinite number of dimensions.  For *any* choice of B_i in a system with finite N (which the authors assume), there will be an infinite number of Poincare recurrences where the information "returns" to the system with arbitrarily high fidelity. Eq. (20) does nothing to change this. It just shows that if you average models with Poincare recurrences at different times then the information transfer looks "permanent" because the information does not return to the system at the same time in all the models.

There are several other places in Sec. III where the author's colloquial statements appear to be making a similar conceptual mistake, but the above example is the most egregious.  They need to fix this issue throughout the manuscript.

(*) The authors write

> Further denoting the probability density function of B_i as P (B_i = x) = f_i(x)...

Since B_i (and hence x) are real numbers, shouldn't f_i be a density?  If no, this should look something like P(B_i \in [x,x+dx]) = f_i(x)dx

(*) The authors give the arXiv identifier for Ref [11] as "arXiv:quant-ph/2208.05497" but the correct identifier is simply "arXiv:2208.05497".  In the past, the arXiv identifier included the topic section (e.g., "quant-ph/xxxxxx"), but this is no longer the case (even though papers are still, of course, sorted by topic on the website).

(*) The authors write

> The destruction of quantum superpositions presupposes a privileged and unique quantum basis.

It seems better to say something like "The destruction of quantum superpositions identifies a privileged and unique quantum basis".  The theory of decoherence doesn't *presuppose* a preferred basis; it derives it from other assumptions, most notably a preferred system.  (It also requires a special form of the Hamiltonian, as the authors investigate, but in principle this is *determined* by the fundamental dynamics of the universe once the preferred system has been identified.)

Relatedly, the authors go on to say

> Any quantum superposition written in this basis decomposes into a classical mixture under the effect of environmental interaction. Thus, pointer states are precisely the only quantum states that are stable under this interaction.

I know the authors know this, but they should make clear that this is a *conclusion* of decoherence theory for *some* Hamiltonians.  The fact that many important systems decohere in a preferred, stable basis *follows* from the dynamics and thereby *defines* the pointer basis; it doesn't arise because one simply chooses to write the state in some presupposed basis.

=== Optional issues ===

(*) The right-hand side of Eq. (A12) is a very unusual and opaque way to express Re \tilde{f}_i(4t)

(*) In eq. (1), why are the authors including the identity operators (for j \neq i) for the single-body term, but not including the identity operators (for k, \neq i,j) for the two-body term?

(*) The authors write

> Mathematically, such an apparatus is described by the pointer observable [Eq. (6)]

This is a little unclear. Do the authors mean that (aspects of) the apparatus are implicitly defined by the fact that it *measures* the pointer observable?  Eq. (6) doesn't really describe the apparatus itself (which would need its own Hilbert space).

Then they say

> By construction, the pointer observable A commutes with the total Hamiltonian (1),

But "by construction" seems wrong.  The only *construction* so far is Eq. (6), and that equation, combined with eq. (1) and/or (2), does not imply that [A,H]=0. I think the authors mean "by definition [of pointer states]" or similar.  (But if that's what they meant, they should explain how the pointer states and pointer observable are related.)

Comments on the Quality of English Language

(*) This was a little hard to parse:

> In a first part of our analysis, we show that the existence of a pointer basis for S imposes a specific structure for the total Hamiltonian describing the evolution of the universe S ⊗ E.

Maybe a colon after "universe" would help?

(*) In the first sentence in Sec. V, "which" should be "that", and there should not be a preceding comma.

(*) In the second-to-last sentence of Sec. III, there should not be a comma after "Both".

Round 2

Reviewer 2 Report

Comments and Suggestions for Authors

I am satisfied with the authors' changes.

Comments on the Quality of English Language

Regarding this sentence

> In the present work we determined the set of qubit models, which support the emergence of classicality.

I said that "which" should be replaced with "that" and the comma dropped.  The authors replied

> This is a stylistic choice and both versions are grammatically correct. 

But this is false.  I'm obviously not going to hold up the publication of the paper because of this, as the editors can deal with minor grammar issues, but the authors should look this stuff up before they reject comments like mine.

In general, "that" is used for restrictive clauses and "which" is used for non-restrictive clauses.  It's true some style guides allow "which" for restrictive clauses, but they *never* allow commas around restrictive clauses.  Commas around restrictive clauses will confuse readers and is not accepted English grammar anywhere.